# Antimicrobial Activity of Different *Artemisia* Essential Oil Formulations

**DOI:** 10.3390/molecules25102390

**Published:** 2020-05-21

**Authors:** Sourav Das, Barbara Vörös-Horváth, Tímea Bencsik, Giuseppe Micalizzi, Luigi Mondello, Györgyi Horváth, Tamás Kőszegi, Aleksandar Széchenyi

**Affiliations:** 1Department of Laboratory Medicine, University of Pécs, Medical School, 7624 Pécs, Ifjúság u. 13., Hungary; pharma.souravdas@gmail.com; 2János Szentágothai Research Center, University of Pécs, Ifjúság u. 20., 7624 Pécs, Hungary; 3Institute of Pharmaceutical Technology and Biopharmacy, University of Pécs, Faculty of Pharmacy, Rókus u. 2., 7624 Pécs, Hungary; barbara.horvath@aok.pte.hu; 4Department of Pharmacognosy, University of Pécs, Faculty of Pharmacy, Rókus u. 2., 7624 Pécs, Hungary; timea.bencsik@aok.pte.hu (T.B.); horvath.gyorgyi@gytk.pte.hu (G.H.); 5Chromaleont s.r.l., c/o Department of Chemical, Biological, Pharmaceutical and Environmental Sciences, University of Messina, 98168 Messina, Italy; giuseppe.micalizzi@chromaleont.it (G.M.); lmondello@unime.it (L.M.); 6Department of Chemical, Biological, Pharmaceutical and Environmental Sciences, University of Messina, 98168 Messina, Italy; 7Unit of Food Science and Nutrition, Department of Medicine, University Campus Bio-Medico of Rome, 00128 Rome, Italy

**Keywords:** *Artemisia* essential oil, Pickering emulsion, oxidative stress, mature biofilm, antimicrobial activity

## Abstract

The extreme lipophilicity of essential oils (EOs) impedes the measurement of their biological actions in an aqueous environment. We formulated oil in water type Pickering *Artemisia*
*annua* EO nanoemulsions (AEP) with surface-modified Stöber silica nanoparticles (20 nm) as the stabilizing agent. The antimicrobial activity of AEP and its effects on mature *Candida* biofilms were compared with those of Tween 80 stabilized emulsion (AET) and ethanolic solution (AEE) of the *Artemisia* EO. The antimicrobial activity was evaluated by using the minimum inhibitory concentrations (MIC_90_) and minimum effective concentrations (MEC_10_) of the compounds. On planktonic bacterial and fungal cells beside growth inhibition, colony formation (CFU/mL), metabolic activity, viability, intracellular ATP/total protein (ATP/TP), along with reactive oxygen species (ROS) were also studied. *Artemisia*
*annua* EO nanoemulsion (AEP) showed significantly higher antimicrobial activity than AET and AEE. *Artemisia*
*annua* EO nanoemulsions (AEP) generated superoxide anion and peroxides-related oxidative stress, which might be the underlying mode of action of the *Artemisia* EO. Unilamellar liposomes, as a cellular model, were used to examine the delivery efficacy of the EO of our tested formulations. We could demonstrate higher effectiveness of AEP in the EO components’ donation compared to AET and AEE. Our data suggest the superiority of the AEP formulation against microbial infections.

## 1. Introduction

*Artemisia annua* L. (Sweet Wormwood, Sweet Annie, Sweet Sagewort, Annual Wormwood, or Qinghaosu) is a very important member of the Asteraceae family, and is widely distributed throughout Asia, South Africa, Europe and North America [1,2,3]. The use of this plant for the treatment of malaria has been recorded before 168 BC in Chinese traditional medicine [2,3,4,5]. Discovery of the artemisinin, a sesquiterpene with the antimalarial property made this plant to have potential commercial importance [4,5,6,7]. The search for other active components has led to the discovery of many phytochemicals including, monoterpenoids, sesquiterpenoids, flavonoids and coumarins, and aliphatic and lipid compounds [6,8]. Apart from its antimalarial activity, the plant has shown anti-tumor, anti-inflammatory, antipyretic, antimicrobial, antiparasitic, antiulcerogenic, and cytotoxic activities [2,3,4,5,6,7,8,9,10]. Several studies on the chemical composition of the *Artemisia annua* L. essential oil (AE) have been performed, and active components like camphor, artemisia ketone, germacrene D, and 1,8-cineole have been found [11,12]. Variability in the chemical composition of AE depending on the geographical origin and plant’s development stages has led to considerable research interests in the investigation of biological properties of AE [3,6]. 

With the increasing demand for the AE in aromatherapy as complementary and as alternative medicine, people believe in the myth that it is harmless; therefore, it has been used for a long time [1]. Previous studies have reported many drug–drug interactions in essential oils [4,13,14]. Several side effects of essential oils, which include allergic reactions, may occur if administered topically. Furthermore, some essential oils can be poisonous if absorbed through the skin, breathed, or swallowed. Uncontrolled widespread usage and continuous production of essential oils as alternative therapy along with carrier diluents have created severe problems, especially among children [13,15,16,17]. Additionally, the highly lipophilic nature of the essential oils makes it difficult to examine their biological properties in aqueous environments [18,19]. Therefore, it is highly important to perform research on the formulations for EOs’ usage, to determine the exact mode of their action, with such knowledge we can suggest appropriate formulation, and avoid the danger of misuse of EOs.

The antimicrobial effectiveness is often described in terms of minimum inhibitory concentration (MIC). Conventional broth dilution techniques and disc diffusion data have provided numerous antimicrobial activities of the AE. However, the diffusion assay is not suitable for essential oil testing, as the components are partitioned through the agar because of their low affinity to water [20,21]. Furthermore, the ability to compare data from the broth and agar dilution methods are limited due to the wide spectrum of test methodologies and selection criteria for end-point determinations [16,22,23]. Several techniques have been used to overcome the lipophilicity of the AE, by the application of suitable solvents such as ethanol, methanol, and dimethyl-sulfoxide (DMSO) or surfactants like Tween 80 (Polyoxyethylene (80) sorbitan monooleate) and Tween 20 (Polyoxyethylene (20) sorbitan monolaurate) [21,24,25]. However, previous studies have repeatedly reported that usage of the solvents/surfactants in various microbiological experiments may contribute to changes in the physicochemical properties of the testing microenvironment, leading to enhancement or reduction of the antimicrobial properties [26,27,28]. Furthermore, the evaporation of the essential oils during the assay or the inability of the active components to reach the test microbes might cause some misleading results [29,30]. 

Thus, novel formulations have been used to enhance the solubility or to emulsify the essential oils in the aqueous environment, resulting in sustained release of the active components in the testing system [29]. The suitability of the organic solvents/surfactants has been questioned, and therefore they are not welcomed in this regard. Efforts to entrap the essential oils by modified cyclodextrins and silica nanoparticles have been made for the precise characterization of the antimicrobial properties [24,31,32].

Approaches to stabilizing the oil-in-water (O/W) and water-in-oil (W/O) emulsions, by the application of solid particles instead of surfactants as stabilizing agents, is novel in the field of essential oil research. The fundamental mechanism involves the adsorption of the solid particles on the oil-water interface, resulting in a significant decrement of the interfacial tension, causing high emulsion stability [21,33]. Decreased evaporation of the essential oils from the nanoparticle-stabilized O/W emulsion formulations, when compared to essential oil-surfactant systems, has been reported previously [34,35]. By application of inert and biocompatible particles instead of surfactants, the irritative and toxic effect of surfactants can be avoided [36].

Even though numerous studies have been performed on the essential oil–Pickering emulsion, literature data on AE (obtained from *Artemisia annua* L. cultivated in Hungary)–nanoparticle formulations related to their application as antimicrobial and anti-biofilm agents are not found [1,3,6,11,14,19]. The primary aim of our present study is to formulate the Pickering nanoemulsion of AE stabilized with surface-modified spherical silica nanoparticles (AEP) and characterize its antimicrobial activity on Gram-positive and Gram-negative bacteria and fungi as well as its effects on mature *Candida* species’ biofilms. We could also demonstrate an effective biofilm-related microbial cytotoxicity and antimicrobial activity of the AEP on planktonic cells and suggest a plausible mode of action of this formulation. Efforts were made in the experiments to support the proposed mode of action.

## 2. Results

### 2.1. Artemisia Essential Oil and Its Components

The amount of the essential oil obtained by steam distillation was 3% *w*/*w* of plant powder. Our gas chromatography–mass spectrometry/flame ionization detection (GC-MS/FID) analyses (Appendix A) has documented β-pinene (1.25%), artemisia ketone (4.43%), yomogi alcohol (1.29%), artemisia alcohol (1.68%), (E)-pinocarveol (7.55%), pinocarvone (3.22%), camphor (7.06%), terpinen-4-ol (1.75%), α-copaene (2.75%), caryophyllene (5.26%), β-farnesene (4.8%), β-selinene (12.27%), spathulenol (1.75%), caryophyllene oxide (8.64%), eudesma-4(15),11-dien-5-ol (1.06%) and mustakone (1.27%) as the major components of the essential oil of *Artemisia annua* L. essential oil, that has been used throughout the experiments.

### 2.2. Preparation and Stability Studies of O/W Type Pickering Nanoemulsions

We found that AEPs and AETs have the same stability. The droplet size of the prepared emulsions increased when the concentration of AE has been raised (Table 1). Since the volume fraction of AE is very low (0.004 or less), we can be sure that the Pickering emulsion is o/w type, as it has been proven before [37].

### 2.3. In Vitro Diffusion Study of Artemisia EO (AE) Formulations

We assumed that the different formulations of AE have different diffusion properties, which could cause different activities in biofilm treatment. To confirm this assumption, in vitro diffusion studies were performed with an ethanolic solution of AE (AEE), in addition to AET and AEP. The concentration of AE was the same for all three formulations, and the droplet sizes of AEP and AET were similar, 160 ± 2.2 and 130 ± 0.9 nm, respectively to avoid the droplet size exclusion effect (Figure 1).

The diffusion profiles were very similar in each case. Still, it can be seen that the cumulative amounts (CA) of AE were the highest in the case of AEP in both experimental circumstances. These were 28.71% at bacterial and 32.30% at fungal experimental temperatures, where AET samples showed lower values, 19.36%, and 30.43%, respectively, while for AEE only 17.60% and 22.82% of AE diffused after 24 h. 

### 2.4. Interaction Studies with Unilamellar Liposomes as Cellular Models

Similar to our previous studies, unilamellar liposomes (ULs) were used as a cellular model for studying the intracellular delivery ability of AE for different formulations [20]. We studied the interaction between 3.5 µm sized ULs and different formulations of AE (AEP, AET, AEE) for 48 h at 30 °C and 24 h at 37 °C. The incubation times and temperatures were the same as in the case of cellular microbiological experiments with fungi and bacteria (Figure 2).

For both experimental circumstances, AEP showed the best ability to deliver AE to the internal water phase of ULs; it was 29.8% after 48 hours at 30 °C, and 59.2% after 24 hours at 37 °C. For the AET samples, these values were 25.0% and 27.3%, respectively, and the lowest values were obtained in the case of AEE samples.

### 2.5. Antibacterial and Antifungal Activities of the Prepared Emulsions

The effects of the *Artemisia* Pickering nanoemulsion, conventional emulsion, and essential oil in ethanol on Gram-positive and Gram-negative bacteria, and on opportunistic fungi were studied. The AEP showed acceptable antibacterial and antifungal activities (MIC_90_) on *E. coli* PMC 201(1.68 ± 0.72 µg/mL), *S. aureus* ATCC 29213 (1.62 ± 0.37 µg/mL), *B. subtilis* SZMC 0209 (1.42 ± 0.64 µg/mL), *P. aeruginosa* PMC 103 (1.46 ± 0.22 µg/mL), *S. pyogenes* SZMC 0119 (3.15 ± 0.16 µg/mL), *S. pombe* ATCC 38366 (2.01 ± 0.46 µg/mL), *C. albicans* SZMC 1372 (3.62 ± 0.65 µg/mL), *C. tropicalis* SZMC 1368 (4.29 ± 0.82 µg/mL), *C. dubliniensis* SZMC 1470 (3.63 ± 0.57 µg/mL) and *C. krusei* SZMC 0779 (3.79 ± 0.57 µg/mL), respectively, when compared to AET (*P* < 0.01). The Pickering *Artemisia annua* EO nanoemulsion (AEP) showed higher antimicrobial activity at an average of twelve-fold less concentration when compared to the free essential oil in ethanol (AEE). The comparative dose–response curves are shown in Figure 3 and Figure 4 for bacteria and fungi, respectively.

### 2.6. Effects of the Minimum Effective Concentration (MEC_10_) on Planktonic Microbial Cells 

The minimum effective concentration (MEC_10_) of the *Artemisia* Pickering nanoemulsion, conventional emulsion and essential oil in ethanol on Gram-positive and Gram-negative bacteria, and on opportunistic fungi are shown in Figure 5 and Figure 6. For the AEP the MEC_10_ concentrations are as follows: *E. coli* (4.05 ± 0.69 µg/mL), *S. aureus* (4.79 ± 0.84 µg/mL), *B. subtilis* (5.54 ± 1.05 µg/mL), *P. aeruginosa* (6.39 ± 0.95 µg/mL), *S. pyogenes* (9.25 ± 1.03 µg/mL), *S. pombe* (7.02 ± 1.55 µg/mL), *C. albicans* (7.12 ± 2.11 µg/mL), *C. tropicalis* (13.79 ± 2.74 µg/mL), *C. dubliniensis* (10.49 ± 3.77 µg/mL) and *C. krusei* (11.67 ± 3.62 µg/mL). The curves expressed a dose-dependent cell survival rate (by CFU quantification) after 60 min exposure to AEP, AET, and AEE. The doses corresponding to MEC_10_ (average 90% survival rate of the ~10^5^ CFU/mL, mid-log phased cell population) were then used in further study on planktonic cells and on mature biofilms. 

### 2.7. Effects on the Microbial Oxidative Balance

Various reactive oxygen species (ROS) production and accumulation in the bacterial and fungal cells initiate oxidative stress followed by cellular structural damages and apoptosis induction [5]. The oxidative stress generation after 60 min of treatment has been investigated (Figure 7 and Figure 8). Data expressed as % of control are as follows: combined ROS detected by DCFDA (1235.46 ± 133.63%), peroxide by DHR 123 (1053.74 ± 146.26%) and superoxide by DHE (1153.84 ± 142.67%) were the highest in the case of *P. aeruginosa*. The AEP caused an effective increase in the combined ROS, peroxide, and superoxide generations as well, in both Gram-positive and negative bacteria and fungi when compared to AET and AEE (*P* < 0.01). The AEE has generated a three to four-fold increment in oxidative stress compared to the growth control (GC), which was the lowest among all treatments. 

### 2.8. Effects on the Microbial Planktonics’ Behavior 

#### 2.8.1. Colony Formation Changes

The changes in the colony formation of the microbial cells were followed using a time-kinetic means of investigating the quantity of the living population after a definite time interval under different samples’ MEC_10_ concentrations (Figure 9 and Figure 10). A significant 50% reduction in the bacterial and fungal cell survivability after 6 and 18 h, respectively, was observed in the case of the AEP exposure when compared to that of AET and AEE (*P* < 0.01). AEE was the least effective among all other treatments. 

#### 2.8.2. Variations in the Intracellular ATP to Total Protein Content Ratios

The energy depletive effects of AEP, AET, and AEE at their MEC_10_ concentrations on the selected Gram-positive and -negative bacteria, and fungi were studied (Figure 11 and Figure 12). Although no significant change in the total protein content (TP) over time in the planktonic cells was observed, a significant 60.37 ± 5.35% decrement in the ATP/TP ratio was found in the case of the AEP treated samples when compared to the 0 h samples (*P* < 0.01). Both AET and AEE have shown a decrease up to an average of 40% in the ATP/TP ratio in the cases of bacteria and fungi when compared to the 0 h samples. 

#### 2.8.3. Effects on Microbial Cell Viability and Metabolic Activity

The multi-parametric cytotoxicity assay has been performed to evaluate the viability of the microbial cells using SYBR green I/PI live-dead cell technique while the resazurin fluorometric method was applied for reflecting the metabolic changes caused by the test samples. The cytotoxic effects of AEP, AET, and AEE on the viability (Figure 13 and Figure 14) and on the metabolic activity (Figure 15 and Figure 16) of the selected bacteria and fungi were evaluated. The 50% reduction in the viable cell population was reached at an average of 9.9 ± 0.71 h and 17.09 ± 0.45 h in the case of the Gram-positive and -negative bacteria, and fungi under AEP exposure. A delayed effect at an average of 9.74 ± 1.65 h was observed in the case of AET to initiate a 50% mean non-viability in the microbial populations when compared to AEP (*P* < 0.01). The AEE has shown the slowest induction of microbial non-viability with a delay of 20.63 ± 3.73 h compared to AEP (*P* < 0.01). A significant 50% decrement in the metabolic activity was observed in the case AEP treated bacteria and fungi at an average of 10.63 ± 0.55 h and 26.35 ± 0.36 h, respectively (Figure 15 and Figure 16). The AET treatment showed an average of 7.42 ± 2.63 h delay in reducing metabolic activity by 50%when compared to AEP treated samples (*P* < 0.01). No significant decrease in the metabolic activity was found in the case of the AEE treated fungal samples.

### 2.9. Effects on Mature Candida Spp. Biofilms

The changes in the metabolic activity, fungal cell viability, total biofilm biomass, total protein content (TP) of the biofilm-attached *Candida* cells, and their ATP content in terms of ATP/TP ratio are shown in Figure 17 and Figure 18. Variable response to the AEP, AET, and AEE treatments at their respective MEC_10_ concentrations was observed when compared to the growth control (GC). Although no changes were found in the total biofilm biomass, a significant 40% reduction in the metabolic activity of the AEP treated mature biofilm-attached *Candida* species after 24 h was observed compared to GC. Whereas, AET and AEE treated samples have shown an average of 20% to 30% metabolic activity reduction compared to the AEP treatment (*P* < 0.01). The reduction of the 50% viable biofilm-attached *Candida* cells was also observed compared to the GC after 24 h of the treatment. The killing activity of the AEE treatments was found to be the lowest compared to the AEP treatments (*P* < 0.01). Total protein (reflecting the number of cells, TP) did not change significantly during the treatments, but a significant 60% reduction of the ATP/TP ratio was found in the case of the AEP treated mature biofilm-attached *Candida* samples when compared to the GC (*P* < 0.01). An average of 28.57 ± 5.26% and 15.34 ± 4.64% reduction in the ATP/TP ratio for the AET and AEE were recorded, resulting in the AEE treatments to be the least effective among the treatment groups. 

## 3. Discussion

The effects of three different formulations of *Artemisia annua* EO on mature *Candida* biofilms and antimicrobial activity have been studied. We succeeded in formulating stable Pickering nanoemulsions using surface-modified silica nanoparticles, and the formulation with desired droplet size (160 nm) was examined for its antimicrobial properties. *Artemisia annua* EO Pickering nanoemulsions (AEP) were stable for three months, with a higher effectiveness when compared to AET and AEE. 

Based on our analyses, AEP has shown stronger antimicrobial activity at lower concentrations (MICs) compared to that of AET and AEE. Several experiments have been conducted previously; however, the mechanism of action at the sub-inhibitory concentrations has not been studied at all [6,12,13]. Our data suggest an effective microbial killing activity of AEP on selected bacteria and fungi. Previous studies believe that the killing action of the essential oils happens due to the leakage in the cells’ cytoplasmic membrane and induction of oxidative stress [38,39,40,41]. We have introduced several staining methods to visualize and to understand the mechanism of action of the AEP, AET and AEE. The AEP was able to induce higher oxidative stress compared to AET and AEE followed by metabolic interference, cell wall disruption, ATP depletion and finally cell death in the case of planktonic bacterial and fungal cells as well as in mature biofilm-attached *Candida* cells at their respective sub-inhibitory concentrations. 

The results obtained from model experiments have highlighted that the Pickering nanoemulsion of the AE oil is the most effective form for the intracellular delivery and for the transport of EO through biofilms when compared to AET and AEE. On the basis of our observations, it can be postulated that the difference in the antibacterial and antifungal activity against microbial cell may be explained by the different adsorption properties of the essential oil formulations [20,34]. Data for unilamellar liposomes as a cellular model support the phenomenon of adsorption of the Pickering emulsion droplets on the cell membrane that has been reported earlier [33]. It might happen that either the passive diffusion occurs due to higher local concentration gradient of the essential oil or fusion of the AEP droplets with the microbial cells allowing the intracellular delivery of the active components from the AEP. The static Franz diffusion experiments on agar gel membrane as a biofilm model showed that AEP has the best ability to deliver the EO through porous structure. The enhanced transportability of AEP must be related to the difference of the surface properties of AEP and AET since experiments were performed with droplets of similar sizes. Overall, our observations demonstrated that the AEP facilitates the *Artemisia* essential oil in penetrating through the biofilm and the cells, inducing oxidative stress and disruption of the cell membrane integrity due to the high adsorption efficacy of Pickering nanoemulsion droplets. 

## 4. Materials and Methods 

### 4.1. Plant Collection and Extraction of Artemisia Essential Oil

The plant samples were collected in 2017 from Békés county, Orosháza-Nagyszénás, Hungary, from a local farm in the autumn and were dried in the dark, under moisture-free conditions for two months. The plant parts were ground to farinose form and were sieved down. The powdered form of the plant was stored in airtight containers for future use. 

The extraction of the *Artemisia annua* L. (*A. annua*) essential oil was done by steam distillation method as described earlier [42]. Briefly, 500 mL of distilled water was added to 50 g dried powder of *A. annua*. The steam distillation was done for 30 min, and the reading in the graduated tube was recorded for the essential oil yield. The yield value was converted to the percentage (%) *w*/*w* of 1 g of the powdered plant’s part. 

### 4.2. GC-MS and GC-FID Studies

The *Artemisia* essential oil was diluted 1:100 in *n*-heptane and injected on the GC-MS and GC-FID systems. The separation and identification of volatile compounds was carried out by using GCMS-QP2020 instrument (Shimadzu, Duisburg, Germany) equipped with a split-splitless injector and an AOC-20i autosampler. For a satisfactory characterization of compounds, two different capillary columns were used. One was a low-polarity column, namely SLB-5ms 30 m × 0.25 mm *id* × 0.25 μm *d_f_* (Merck Life Science, Rome, Italy), while the other with a medium polarity was a Supelcowax-10 30 m × 0.25 mm *id* × 0.25 μm *d_f_* (Merck Life Science). Both columns operated under a programmed temperature: 50 °C to 280 °C at 3.0 °C/min. Injection volume was of 0.5 µL with a split ratio of 1:10. Helium was used as gas carrier at a constant linear velocity of 30 cm/s and an inlet pressure of 26.7 kPa. MS parameters were as follows: the mass range was 40–650 amu, the ion source temperature was 220 °C and the interface temperature was 250 °C. The GCMSsolution software (version 4.50 Shimadzu, Kioto, Japan) was used for data collection and handling. Peak identification was carried out by using a double parameter: MS spectra similarity (over 85%) and a LRIs ±5 and ±10 linear retention index (LRI) tolerance window for the SLB-5ms and Supelcowax-10 column, respectively. In this respect, a homologous series C7-C30 *n*-alkanes (Merck Life Science) standard mixture in hexane (1000 g/mL) was used for LRIs calculation on SLB-5ms column, while a C4-C24 even carbon saturated FAMEs (Merck Life Science, Rome, Italy) standard solution in hexane (1000 g/mL) on Supelcowax-10 column. For mass spectral identification, *FFNSC 3.01* (Shimadzu Europe, Duisburg, Germany) was mainly used.

The quantification of volatile compounds was carried out by using a GC-2010 Plus (Shimadzu, Kioto, Japan) equipped a split-splitless injector (280 °C), an AOC-20i autosampler and an FID detector. The GC columns, temperature program, and carrier gas were the same as described for the GC-MS system, except the initial inlet pressure (99.5 kPa) (the average linear velocity was 30 cm/s).The FID temperature was set at 280 °C (sampling rate: 40 ms), while the gas flows were 40 mL/min for hydrogen, 30 mL/min for the make-up gas (nitrogen) and 400 mL/min for air. Data were collected and processed through the LabSolution software (version 5.92, Shimadzu). 

### 4.3. Preparation and Characterization of Pickering Nanoemulsion

We synthesized surface-modified spherical silica nanoparticles (d = 20 nm) using the Stöber method [43] and used them as a stabilizing agent of Pickering nanoemulsions. For the stabilization of Pickering nanoemulsions, silica nanoparticles were synthesized as described previously [20]. The surface of hydrophilic silica nanoparticles was modified with propyltriethoxysilane (PTES Alfa Aesar, Haverhill, MA, USA, pur. 99%) with a theoretical surface coverage of 20%. The size distribution of nanoparticles was determined by dynamic light scattering measurements (DLS, Malvern Zetasizer Nano S, Malvern Panalytical Ltd., Worcester, UK), the particles were suspended in ethanol or water. The following data and settings were used in DLS measurements: temperature 25 °C; disposable polycarbonate cuvette; autocorrelation function; count rate between 200-500 kcps; refractive index and absorption of silica of 1.57 and 0.001, respectively; refractive index of ethanol and water of 1.367 and 1.436, respectively; duration 60 s. Morphology was examined with transmission electron microscopy (TEM, JEM-1400, JEOL Ltd., Tokyo, Japan). The drop of the sample suspension (~1 µL) was drop cast onto 200 mesh copper grid coated with carbon film (EMR Carbon support grids, Micro to Nano Ltd., Haarlem, The Netherlands) and dried overnight in vacuum desiccators. The TEM images of surface modified silica nanoparticles can be seen in Appendix A.

For the Pickering emulsions (AEP), the concentration of nanoparticles was kept constant (1 mg/mL) and concentration of *Artemisia* essential oil (AE) was varied between 200 µg/mL and 3.5 mg/mL. The emulsification process was carried out in two steps. In the first step, the mixture was sonicated for 2 min (Bandelin Sonorex RK 52H, BANDELIN electronic GmbH & Co. KG, Berlin Germany). The second step was high shear emulsification, which was performed with an UltraTurrax (IKA Werke T-25 basic, IKA^®^-Werke GmbH & Co. KG, Germany) high shear homogenizer for 2 min at 13,500 rpm. Conventional emulsions, stabilized by Tween 80 surfactant (AET) (Acros Organics, Thermo Fisher Scientific, Waltham, MA, USA) and ethanolic solution (AEE) were also prepared to compare stability parameters, droplet sizes and microbiological properties with those of AEP. The concentration of Tween 80 surfactant was the same as nanoparticles, 1 mg/mL. The stability of emulsions was studied based on periodical droplet size determination using DLS measurements (Malvern Zetasizer Nano S, Malvern Panalytical Ltd., Worcester, UK). The emulsions were stored in airtight vials, covered with aluminum foil at room temperature (25 °C). We considered the emulsion to be stable when the droplet size did not change in time, creaming, phase separation, or sedimentation and aggregation of nanoparticles could not be observed.

### 4.4. In Vitro Diffusion Studies—Static Franz Diffusion Cell Method

We carried out the diffusion study with the same parameters as in the in vitro cellular experiments—30 °C/24 h for fungi and 37 °C/24 h for bacteria—to obtain data that could be used to compare with the antimicrobial activities. The examination of diffusion properties was performed in static vertical Franz diffusion cells (Hanson Microette Plus, Hanson Research 60-301-106, Hanson Research Corporation, Chatsworth CA, USA). All experiments were performed in triplicate. As a biofilm model membrane, we have used 2.1 mm thick 2 *w*/*w*% agar gel membranes with effective penetration area of 2.54 cm^2^, with a pore size of 1.5-3 µm, as its tortuous pore structure can model the biofilm matrix [44,45]. The agar membrane was prepared as described previously [34]. The volume of the receiver chamber was 7 mL; the receiver solution was phosphate buffered saline (PBS). For PBS preparation the following salts were used: NaCl (high purity, VWR Chemicals Ltd., Debrecen Hungary), KCl (purity 99%–100.5%, VWR Chemicals Ltd., Debrecen Hungary), Na_2_HPO_4_∙2H_2_O (AnalaR NORMAPUR^®^, purity ≥9 9.0%, VWR Chemicals Ltd., Debrecen Hungary) and KH_2_PO_4_ (purity ≥ 99.0%, VWR Chemicals Ltd., Debrecen Hungary). The volume of emulsion or solution sample was placed in the donor chamber (600 µL), and the diffusion was examined for 24 h, the samples were collected at 15 min, 30 min, 1 h, 2 h, 3 h, 6 h and 24 h. The volume of the taken sample was replaced with fresh PBS buffer. The (AE) content was determined with UV-Vis spectroscopy (Jasco V-670 UV/VIS Spectrophotometer, ABL&E-JASCO Ltd., Budapest, Hungary). The absorption wavelength of *Artemisia* essential oil was 248 nm.

### 4.5. Interaction Study between the Cellular Model (Unilamellar Liposomes) and Different Formulations of AE

Unilamellar liposomes (ULs) were prepared from phosphatidylcholine (Phospolipon 90G, Phospholipid GmbH, Berlin, Germany) as described previously [20,46]. A 5 mL suspension of ULs was mixed with 3 mL Pickering nanoemulsion, conventional emulsion, or ethanolic solution, and the *Artemisia* essential concentration was 200 μg/mL for all examined formulations. In the first set of experiments, the mixture was stirred at 600 rpm for 24 h at 37 °C, and 1 mL aliquots were taken after 1, 6, and 24 h. In the second set of experiments, the mixture was stirred at 600 rpm for 48 h at 30 °C, and 1 mL aliquots were taken after 1, 6, 24, 30 and 48 h. The samples were centrifuged at 3000 rpm and 20 °C for 5 min, and the ULs were collected and dissolved in absolute ethanol. The *Artemisia* EO content of samples was determined with UV/Vis Spectroscopy at 248 nm (Jasco V-550 UV/VIS Spectrophotometer; ABL&E-JASCO Ltd., Budapest, Hungary). For UV/Vis measurements, we prepared samples without *Artemisia* essential oil, i.e., ULs with silica nanoparticle suspension, Tween 80 solution, or ethanol were also mixed and centrifuged and were used as blanks.

### 4.6. Materials and Microorganisms Used for the Biological Experiments 

Promega BacTiter-Glo microbial cell viability assay kit (Bio-Science, Budapest, Hungary), glass beads (Sigma-Aldrich Chemie GmbH, Steinheim, Germany, reference number: G-8772), sterile petri-dishes (Greiner Bio-One, Kremsmunster, Austria), sterile petri-dishes for the biofilm assays (Sarstedt AG & Co. KG, Numbrecht, Germany, reference number: 83.3900.500), 0.22 µm vacuum filters (Millipore, France), cell scraper (Sarstedt AG & Co. KG, Numbrecht, Germany), sterile microplates (Greiner Bio-One, Kremsmunster, Austria), sterile microplates for the biofilm assays (Sarstedt AG & Co. KG, Numbrecht, Germany, catalog number: 83.3924.500), 96-well optiplates (Perkin Elmer, Waltham, Massachusetts, USA), bovine serum albumin (BSA; Biosera, Nuaille, France), potassium phosphate monobasic, ethanol 96% (Et), methanol, peptone, yeast extract, agar-agar and Mueller-Hinton agar (for the maintenance of the tested bacteria’s health, used throughout the experiments) (Reanal Labor, Budapest, Hungary), modified RPMI 1640 (contains 3.4 *w*/*v*% MOPS, 1.8 *w*/*v*% glucose and 0.002 *w*/*v*% adenine), menadione (ME) (Sigma Aldrich, Budapest, Hungary), disodium phosphate, dimethyl-sulfoxide (DMSO) from Chemolab Ltd. (Budapest, Hungary), sodium chloride (VWR International Ltd., Debrecen, Hungary), potassium chloride (Scharlau Chemie S.A, Bercelona, Spain), 3-(N-morpholino) propanesulfonic acid (MOPS) (Serva Electrophoresis GmbH, Heidelberg, Germany), Caspofungin (CAS) from Merck Sharp & Dohme Ltd., Netherlands, vancomycin (VAN) from Fresenius Kabi Ltd., (Budapest, Hungary), SYBR green I 10,000×, propidium iodide, dihydrorhodamine 123 (DHR 123), 2′,7′–dichlorofluorescin diacetate (DCFDA) and dihydroethidine (DHE) were purchased from Sigma Aldrich (Budapest, Hungary) and from the above-mentioned sources. All other chemicals applied for the study were of analytical or spectroscopic grade. For fungi, we used an in-house nutrient medium containing 1 *w*/*v*% yeast extract, 2 *w*/*v*% peptone, 2 *w*/*v*% glucose and 2 *w*/*v*% agar–agar (for the colony-forming unit assay in the petri-dishes) [47], while phosphate-buffered saline (PBS, pH 7.4) was obtained from Life Technologies Ltd., Budapest, Hungary. Highly purified water (<1.0 µS) was applied throughout the studies.

*Escherichia coli* (*E. coli*) PMC 201, *Pseudomonas aeruginosa* (*P. aeruginosa*) PMC 103, *Bacillus subtilis* (*B. subtilis*) SZMC 0209, *Staphylococcus aureus* (*S. aureus*) ATCC 29213, *Streptococcus pyogenes* (*S. pyogenes*) SZMC 0119, *Schizosaccharomyces pombe* (*S. pombe*) ATCC 38366, *Candida albicans* (*C. albicans*) SZMC 1372, *Candida tropicalis* (*C. tropicalis*) SZMC 1368, *Candida dubliniensis* (*C. dubliniensis*) SZMC 1470 and *Candida krusei* (*C. krusei*) SZMC 0779 were obtained from Szeged Microbial Collection, Department of Microbiology, University of Szeged, Hungary (SZMC) and Department of General and Environmental Microbiology, Institute of Biology, University of Pecs, Hungary (PMC).

### 4.7. Determination of Minimum Inhibitory Concentration (MIC_90_)

We used a previously published protocol [4,20] for measuring the antibacterial activity separately on *E. coli*, *P. aeruginosa*, *B. subtilis*, *S. aureus*, and S. *pyogenes* and antifungal activity against *S. pombe*, and *Candida* species. In brief, for the antibacterial activity determination, a bacterial population of ~10^5^ CFU/mL was inoculated in modified RPMI 1640 medium followed by incubation for 16 h at 35 ± 2 °C Thermo Scientific Heraeus B12 microbiological incubator (Auro-Science Consulting Kft., Budapest, Hungary) with AEP, AET, AEE and VAN over a wide range of concentrations (26.2—0.01 µg/mL). The absorbance was measured by a Thermo Scientific Multiskan EX 355 microplate reader (InterLabsystems, Budapest, Hungary) at 600 nm.

The antifungal activity against *S. pombe* and *Candida* species were also performed according to our previously published method [20]. Briefly, ~10^3^ cells/mL were incubated with AEP, AEE, AET and CAS at a wide concentration range (26.2–0.01 µg/mL) in modified RPMI 1640 medium for 48 h at 30 °C (Sanyo MIR-154 microbiological incubator, Auro-Science Consulting Kft., Budapest, Hungary). The absorbance values obtained by the microplate reader at 595 nm were converted to percentages and were compared to the growth control (100%). The data were fitted by a non-linear dose–response curve fitting method to estimate the dose producing ≥90% growth inhibition (MIC_90_). All measurements were performed by applying three technical replicates in six independent experiments. VAN and CAS were used as the standard antibacterial and antifungal drug controls, respectively throughout the experiments. 

### 4.8. Determination of Minimum Effective Concentration (MEC_10_) 

The MEC_10_ concentration was obtained according to our previously published protocol [4]. In brief, a wide concentration range (105–0.2 µg/mL) of AEP, AET and AEE was used to treat mid-log phased ~10^5^ cells/mL in modified RPMI 1640 medium for an hour, incubated at 35 ± 2 °C and 30 °C for the bacteria and for the fungi, respectively, in an orbital incubator (Sanyo MIR-220RU orbital incubator, Auro-Science Consulting Kft., Budapest, Hungary). Inoculated growth medium without any treatment was considered as growth control. For the colony-forming unit (CFU/mL) quantification, 1 mL of the treated and untreated samples were pipetted out and 10^5^ times dilutions were prepared followed by spreading 50 µL of the test samples onto 20 mL respective nutrient-rich agar media containing petri-dishes and were incubated for 24 h at 35 ± 2 °C and 30 °C in the case of bacteria and fungi, respectively. The colony formation data (CFU/mL) were converted to percentages and the data were fitted using a non-linear dose–response curve fitting method to evaluate the drug concentrations producing approximately 90% microbial cell population growth (MEC_10_) when compared to the untreated microbial cell populations after one hour of treatment. Three technical replicates in six independent experiments were performed in all the measurements. VAN and CAS were used as the standard antibacterial and antifungal drug controls throughout the experiments, respectively. 

### 4.9. Quantification of Microbial Oxidative Stress Production in the Planktonic Cells

#### 4.9.1. Combined ROS Generation Measurement

For the quantification of the combined (not specified separately) reactive oxygen species (ROS) generation, we have followed our previous protocol [4,20]. In brief, for the ROS measurements, mid-log phased ~10^5^ cells/mL were collected and centrifuged at 1500 *g* (Hettich Rotina 420R bench-top centrifuge, Auro-Science Consulting Ltd., Budapest, Hungary) for 5 min and were re-suspended in PBS. The staining of the cells was done with 2′,7′–dichlorofluorescein diacetate (DCFDA) stock solution (20 mM in DMSO) with a final concentration of 25 µM in each well, followed by incubation at 35 ± 2 °C for the bacteria and 30 °C for the fungi in the dark with mild rotation in a test tube rotator (Cole-Parmer Roto-Torque Variable Speed Rotator, InterLabsystems, Budapest, Hungary) for 30 min. The cells were re-suspended in modified RPMI 1640 medium after centrifugation followed by treatment with AEP, AET and AEE at their respective MEC_10_ concentrations for one hour. The fluorescence data were recorded by a fluorescence spectrophotometer/microplate reader (Hitachi F-7000 fluorescent spectrophotometer, Auro-Science Consulting Ltd., Budapest, Hungary) at Ex/Em = 485/535 nm in 96-well optiplates. VAN and CAS were used as the standard antimicrobial controls in the case of bacteria and fungi. ME treated samples were considered as the positive controls. The percentage increase (%) in the ROS generation was measured by comparing the signals obtained from the growth controls (GC). Six independent experiments with three technical replicates in each treatment were performed. 

#### 4.9.2. Detection of Peroxide (O_2_^2-^) and Superoxide Anion (O_2_^•-^) Generation

Our previously described protocol was adapted for the peroxide and the superoxide anion radicals’ measurements [20,47]. AEP, AET and AEE at their respective MEC_10_ concentrations were used to treat the mid-log phased (~10^5^ cells/mL) microbial cell suspensions followed by incubation in an orbital incubator at 35 ± 2 °C and 30 °C for the bacteria and fungi in modified RPMI 1640 medium for an hour, respectively. VAN and CAS were used as the standard antimicrobial controls in the case of bacteria and fungi. Menadione (ME) (0.5 mmol/L, end concentration) was used as the positive control throughout the experiments. The cells were centrifuged at 1500 *g* for 5 min at room temperature and were re-suspended in PBS at the original volume of the samples. DHR 123 (10 µmol/L end concentration) and DHE (15 µmol/L) were used to stain the microbial cell samples for the peroxide and superoxide determination and were further incubated at 35 ± 2 °C for the bacteria and 30 °C for the fungi in the dark with mild shaking. The samples were centrifuged again and were re-suspended in PBS, followed by even sample distribution into the 96-well microplates. For the peroxide measurements, the fluorescence was measured at Ex/Em: 500/521 nm, whereas, Ex/Em: 473/521 nm was used to detect the superoxide generation by a fluorescence spectrophotometer/microplate reader in 96-well optiplates. The percentage increase (%) in the oxidative stress was measured by comparing the signals obtained from the growth controls (GC). Six independent experiments were performed with three technical replicates for each treatment. 

### 4.10. Microbial Cytotoxicity and Viability Kinetic Assays of Planktonics

The cytotoxicity and the viability of the microbial cells were performed with our previously published method [4,20]. Briefly, AEP, AET, AEE, VAN (bacteria) and CAS (fungi) at their respective MEC_10_ concentrations were used to treat the mid-log phased microbial cell populations (~10^5^ cells/mL) in modified RPMI 1640 medium followed by incubation in an orbital shaker-incubator at 35 ± 2 °C and 30 °C for the bacteria and fungi in modified RPMI 1640 medium. After that, the multi-parametric cytotoxicity and viability kinetic studies were performed (Section 4.10.1, Section 4.10.2, Section 4.10.3 and Section 4.10.4.). 

#### 4.10.1. Determination of the Colony-Forming Unit (CFU/mL) 

We followed our previously published protocol for determining the changes in the colony formation (CFU/mL) [4]. Briefly, 1 mL of the treated and the untreated samples were pipetted at the 0, 2, 6, 8, 16 and 24 h for the bacteria, and 0, 6, 12, 30, 36 and 48 h for the fungi, and were diluted 10^5^ times followed by spreading 50 µL onto 20 mL nutrient agar media with a cell spreader and incubation at 35 ± 2 °C (bacteria) and 30 °C (fungi) for 24 h. VAN and CAS were used as the reference controls for the bacteria and fungi. Controls prior to the treatment at 0 h were considered to be 100% and sigmoidal time interval vs. % CFU decrease was established for each test samples. Three technical replicates for each treatment in six independent experiments were performed.

#### 4.10.2. Quantification of the Planktonics’ Intracellular ATP and Total Protein Contents

The instructions provided by the manufacturer was followed for the intracellular ATP measurement of the microbial planktonics. In brief, 1 mL of both treated and untreated aliquots were sampled at time intervals of 0, 2, 6, 8, 16 and 24 h (for bacteria), and 0, 6, 12, 30, 36 and 48 h (for fungi) and were centrifuged at 1000 *g* for 5 min. The pellets were washed and were re-suspended in 1 mL PBS. Equal volume of the freshly prepared BacTiter-Glo reagent was pipetted to 100 µL of PBS containing the planktonic cells in 96-well optiplates. The microplates were shaken in the dark at 4 °C for 15 min on an orbital shaker and the luminescence was measured using a Perkin Elmer EnSpire Multimode plate reader (Perkin Elmer, Waltham, MA, USA). All the measurements were calculated in terms of nmol/L from an ATP standard calibration curve and were converted to intracellular ATP over total protein content ratio (nmol/mg) followed by comparison with the controls prior to the treatment at 0 h. The PBS, VAN (for bacteria) and CAS (for fungi) were considered as the blank and standard antimicrobial controls throughout the experiments. Three technical replicates were performed for each treatment in six independent experiments [48].

For the total protein content of the planktonics, we adapted the Bradford protein assay method [49]. One mL of AEP, AET and AEE treated samples were collected at time intervals of 0, 2, 6, 8, 16 and 24 h (for bacteria), and 0, 6, 12, 30, 36 and 48 h (for fungi) and were centrifuged at 1000 *g* for 5 min. The pellets were washed and were re-suspended in 1 mL NaOH (1 M) containing 100 mg of glass beads (425-600 µm) [50,51]. The collected samples were stabilized on ice for 5 min followed by 5 min vortexing with a mechanical cell disruptor (Scientific Industries SI analog cell disruptor (InterLabsystems, Budapest, Hungary), consecutively. The cycle was repeated three times and then the samples were centrifuged at 20,000 *g* for 10 min at 4 °C (Heraeus Multifuge 3 S-R bench-top centrifuge, Kendo Laboratory products, Osterode, Germany). The supernatants’ microbial cell lysates were collected in separate 1.5 mL centrifuge tubes. Two hundred microliters of the freshly prepared Bradford reagent was added to a new 96-well general microplate containing 20 µL of the test samples lysates. The absorbance was measured at 595 nm by a multimode plate reader. NaOH, VAN and CAS treated samples were used as the blank, standard antibacterial and antifungal controls, respectively. Three technical replicates for each treatment in six independent experiments were performed. All the measurements were calculated in terms of mg/L from a BSA standard calibration curve and were compared to the 0 h samples [48].

#### 4.10.3. Quantification of the Planktonics’ Metabolic Activity

For the metabolic activity measurements, AEP, AET and AEE treated samples were pipetted at time intervals of 0, 2, 6, 8, 16 and 24 h (for bacteria), and 0, 6, 12, 30, 36 and 48 h (for fungi) and were centrifuged at 1000 *g* for 5 min. The pellets were washed and were re-suspended in 200 µL PBS containing resazurin (12.5 µmol/L) followed by 40 min of incubation at 30 °C. The fluorescence was measured with a fluorescence spectrophotometer/microplate reader in 96-well optiplates at an Ex/Em: 560/590 nm wavelengths. The percentage (%) metabolic activity was estimated based on the fluorescence intensities of samples prior to the application of the treatment at 0 h which were considered to be 100%. A sigmoidal time interval vs % metabolic activity was established for each test samples. The PBS, VAN (for bacteria) and CAS (for fungi) were considered as the blank and standard antimicrobial controls throughout the experiments. Three technical replicates for each treatment in six independent experiments were performed [4].

#### 4.10.4. Live/Dead Discrimination of the Planktonic Microbial Cells

Briefly, both untreated, AEP, AET and AEE treated samples were taken at 0, 2, 6, 8, 16 and 24 h (for bacteria), and 0, 6, 12, 30, 36 and 48 h (for fungi) following centrifugation at 1000 *g* for 5 min, washed and re-suspended in PBS (100 µL/mL). Freshly prepared 100 µL of the SYBR Green I/PI working solutions [4] were added to the samples. For 15 min, the plates were incubated in the dark with mild shaking at room temperature. Fluorescence intensities of the SYBR Green I and PI were measured in 96-well optiplates at an Ex/Em: 490/525 nm and 530/620 nm wavelengths, respectively, by a fluorescence spectrophotometer/microplate reader. The green to red fluorescence ratio for each sample and each dose was established and the % of the dead (non-viable) microbial cells with the response to the applied dose was plotted against the applied test compounds using our previous formula [20]. The VAN and CAS were considered as the standard antimicrobial controls, whereas, PBS was used as the blank. All the treatments were done in triplicates and six independent experiments were performed. 

### 4.11. Determination of the Effects on Preformed Mature Candida Biofilms

The effects (Section 4.10.1, Section 4.10.2, Section 4.10.3 and Section 4.10.4) on the mature *Candida* species biofilms and the treatment conditions were adapted from our previously published work [4]. Briefly, 200 µL of 24 h-old late-log phased (~10^6^ cells/mL) *C. albicans* and non-*albicans* species in modified RPMI 1640 medium were used to culture biofilms in the microplates prior to the treatments for 24 h at 30 ºC. The microplates were washed in PBS followed by re-incubation in 200 µL modified RPMI 1640 medium containing AEP, AET and AEE at MEC_10_ concentrations (µg/mL) for further 24 h at 30 °C. Modified RPMI 1640 medium, untreated growth sample and CAS treated samples were considered as the blank, growth control (GC) and standard antimicrobial control (CAS), respectively. The biofilm degradation percentage (%) was measured based on the comparison of the values with those of the GC. All the treatments were performed in three technical replicates in six independent experiments. 

#### 4.11.1. Evaluation of the Total Fungal Biomass

The change in the biofilm biomass was determined by our previously published crystal violet assay [4]. After 24 h of incubation, the supernatant containing the test samples was pipetted out and the wells were washed with PBS. Two hundred microliters of methanol was added to each well in order to fix the biofilms. The supernatant was removed and 200 µL of 0.1% crystal violet in absolute ethanol was pipetted into each well. The crystal violet solution was pipetted out and 200 µL of acetic acid (33% v/v in double distilled water working stock) was pipetted to the crystal violet pre-stained biofilms to release the bound stains. Subsequently, 20 min later, the acetic acid-dissolved dye from the stained biofilms was pipetted into the wells of general microplates. The absorbance was measured at 590 nm by a multimode plate reader. The CAS was considered as the standard antifungal control. The change in the percentage (%) biofilm biomass was evaluated from the growth control (GC) values which were taken as 100% fungal biofilm biomass. Three technical replicates in each treatment for six independent experiments were considered. 

#### 4.11.2. Metabolic Activity in the Biofilms

A resazurin-derived fluorescent technique was adapted from our previously published protocol to determine the metabolic activity in the fungal biofilms [4]. Briefly, after 24 h of the treatment with AEP, AET and AEE at their respective MEC_10_ concentrations, all of the supernatants were removed followed by rinsing the wells with the PBS. Two hundred microliters of sterile PBS containing resazurin (12.5 µmol/mL) was pipetted into the wells and was incubated at 30 °C for 40 min. The fluorescence intensities were recorded at an Ex/Em: 560/590 nm wavelengths, respectively, by a fluorescent spectrophotometer/microplate reader. The percentage (%) metabolic activity measurements were evaluated based on the fluorescence values recorded by the growth control (GC), which was considered as 100%. The PBS was taken as the blank throughout the measurements. Six independent experiments were performed with three technical replicates for each treatment. 

#### 4.11.3. Viability Assay of the Biofilms

For the discrimination of the live/dead fungal cells in the biofilms, we have followed a method previously published [4,17]. After 24 h of treatment with AEP, AET and AEE, at their respective MEC_10_ concentrations in modified RPMI 1640 medium, the planktonic cells were pipetted out, followed by rinsing and filling of the biofilm-attached wells with 100 µL PBS. One hundred microliters of SYBR Green I/PI working solutions [17] were pipetted into the wells of the microplate. The plates were incubated at room temperature for 15 min in the dark with mild shaking. A fluorescence spectrophotometer/microplate reader was used to measure the fluorescent intensities at an Ex/Em: 490/525 nm and 530/620 nm for SYBR Green I and PI, respectively, into 96-well optiplates. The green to red fluorescence ratio for each sample and each dose was established, and the % of the dead microbial cells with the response to the applied dose was plotted against the applied test compounds using our previous formula [17]. All treatments were done using three technical replicates in six independent experiments. 

#### 4.11.4. Determination of the Candida Biofilms’ Intracellular ATP and Total Protein Contents

The instructions provided by the company was followed for the intracellular ATP measurement of the *Candida* mature biofilms. In brief, after 24 h of the treatment with AEP, AET and AEE, the planktonic cells were pipetted out, followed by rinsing the biofilm and pipetting 100 µL of PBS into every well of the microplate. An equal volume of the freshly prepared BacTiter-Glo reagent was pipetted into the wells containing the biofilms submerged in the PBS. The microplates were shaken in the dark at 4 °C for 15 min on an orbital shaker. The content of the microplates was transferred to 96-well optiplates and the luminescence was measured using a multimode plate reader. All measurements were quantified by using an ATP standard calibration curve and were converted to intracellular ATP over total protein content ratio (nmol/mg) followed by comparison with the growth control (GC). The PBS and CAS were used as the blank and the standard antifungal controls, respectively. Three technical replicates for each treatment in six independent experiments were performed [49]. 

For the measurement of the total protein content in the biofilms, parallel to the microplate cultures the biofilms were produced and treated in 35 mm diameter sterile petri dishes with identical surface properties to those of the microplates. We have adapted the Bradford protein assay method [49]. Briefly, after 24-hour treatment of the mature biofilms with AEP, AET and AEE, the planktonic cells were pipetted out and biofilms were rinsed with PBS. The biofilms were scraped out from the petri dishes using a cell scraper into sterile 1.5 mL centrifuge tubes containing 1 mL NaOH (1 M) and 100 mg of glass beads (425–600 µm). The collected samples were stabilized on ice for 5 min, followed by 5 min vortexing, consecutively. The cycle was repeated three times and then the samples were centrifuged at 20,000 *g* for 10 min at 4 ºC. The supernatants’ fungal cell lysates were pipetted out and were diluted in NaOH (1 M) up to the initial volume used for the biofilm growth. Two hundred microliters of the homemade freshly prepared Bradford reagent were added to a new 96-well general microplate containing 20 µL of the test sample lysates. The absorbance was measured at 595 nm by a multimode plate reader. All the measurements were calculated in terms of mg/L from a BSA standard calibration curve. Total protein contents were referred to the surface are of the microplates in order to obtain the valid ATP/TP values. The NaOH and CAS treated samples were used as the blank and the standard antifungal controls, respectively. Three technical replicates for each treatment in six independent experiments were performed. All the measurements were calculated in terms of mg/L from a BSA standard calibration curve and were compared to the growth control (GC).

### 4.12. Stastical Analyses

All data are given as mean ± SD. Graphs and statistical analyses were conducted using OriginPro 2016 (OriginLab Corp., Northampton, MA, USA). All experiments were performed independently six times in triplicates and data were analyzed by one-way ANOVA test. *P*-values of <0.05 were considered statistically significant. The minimum inhibitory concentration (MIC_90_) and the minimum effective concentration (MEC_10_) were calculated using a non-linear dose–response curve function as follows:(1)$$y=A_{1}+\frac{A_{2}-A_{1}}{1+10^{(LOG_{x}0-x)p}}$$
where A_1_, A_2_, LOG_x_0, and p are the bottom asymptote, top asymptote, center, and hill slope of the curve have been considered.

## Figures and Tables

**Figure 1 molecules-25-02390-f001:**
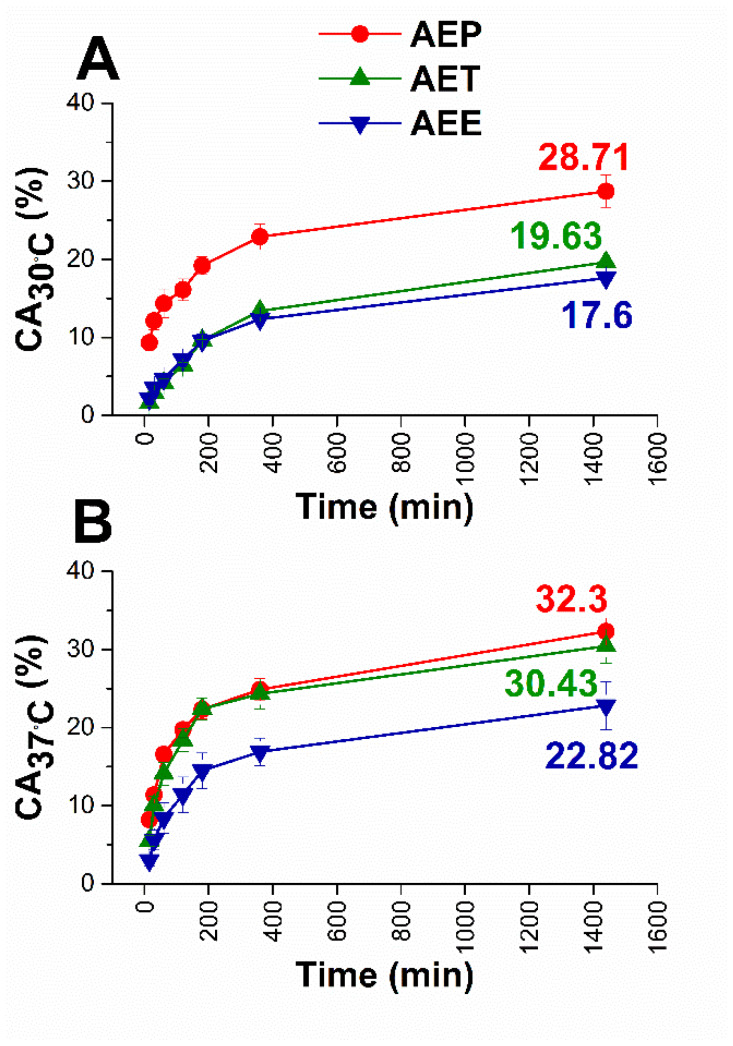
Results of in vitro diffusion studies through agar gel membrane as biofilm model for AEE, AET, and AEP at 30 °C (**A**) and 37 °C (**B**), respectively. The AE concentration is 200 µg/mL. CA: cumulative amounts of essential oil in percentage; the experiments were performed in triplicate (mean ± SD).

**Figure 2 molecules-25-02390-f002:**
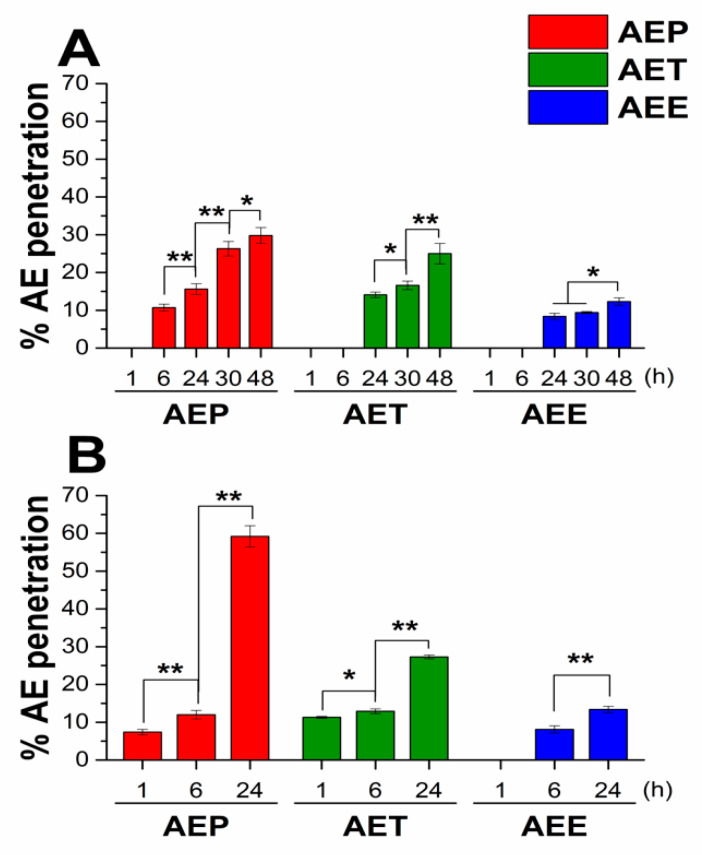
Results of the interaction study between unilamellar liposomes (Uls) and different formulations of *Artemisia* essential oil (AE) at 30 °C for 48 h (**A**) and at 37 °C for 24 h (**B**), experiments were performed in triplicate. AEP: Pickering emulsion, AET: conventional emulsion, AEE: ethanolic solution. c[AE] = 200 µg/mL (** P < 0.05* and *** P < 0.01*, mean ± SD).

**Figure 3 molecules-25-02390-f003:**
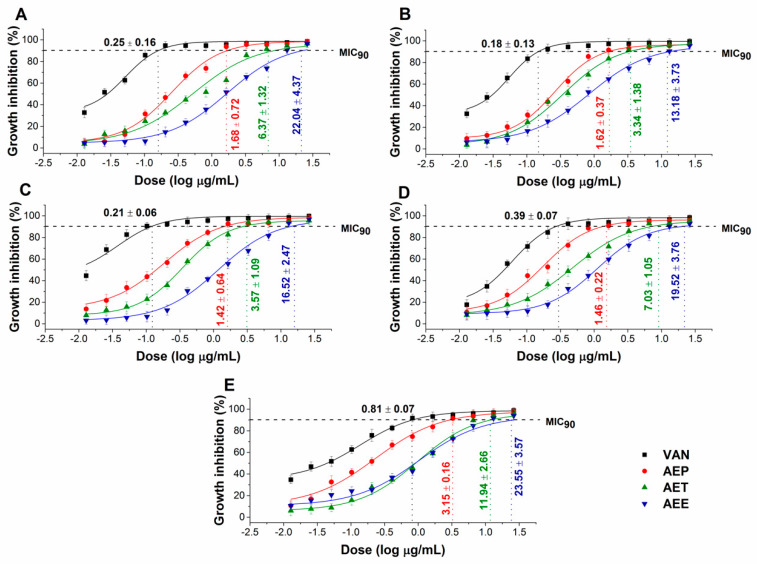
Minimum inhibitory concentration (MIC_90_) of AEP, AET, AEE, and vancomycin (VAN) in µg/mL on *E. coli* (**A**), *S. aureus* (**B**), *B. subtilis* (**C**), *P. aeruginosa* (**D**), and *S. pyogenes* (**E**). Six independent experiments each with three technical replicates were performed (mean ± SD).

**Figure 4 molecules-25-02390-f004:**
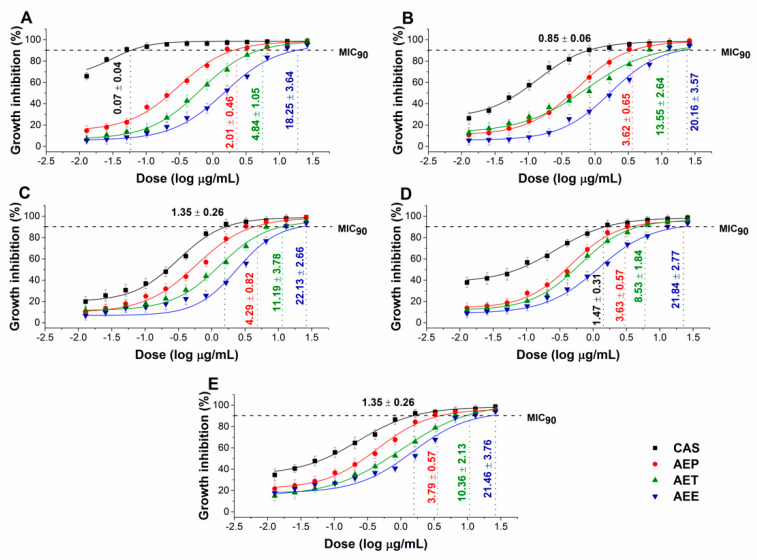
Minimum inhibitory concentration (MIC_90_) of AEP, AET, AEE, and caspofungin (CAS) in µg/mL on *S. pombe* (**A**), *C. albicans* (**B**), *C. tropicalis* (**C**), *C. dubliniensis* (**D**) and *C. krusei* (**E**). Six independent experiments each with three technical replicates were performed (mean ± SD).

**Figure 5 molecules-25-02390-f005:**
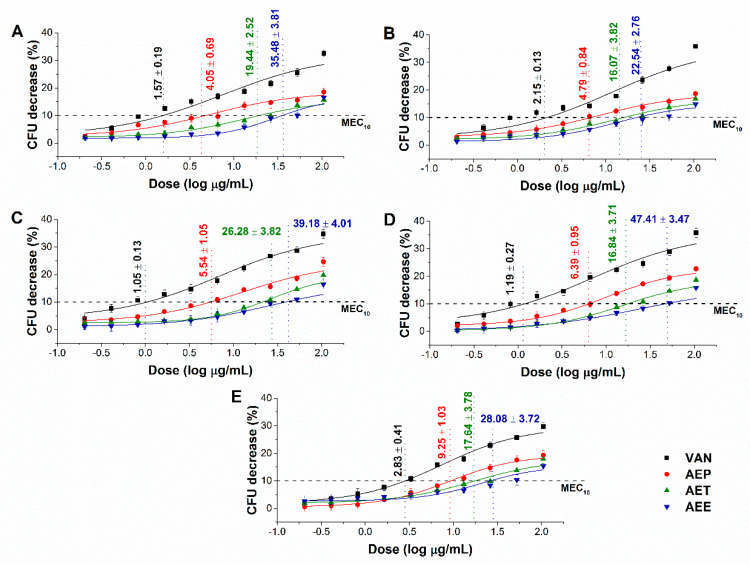
Minimum effective concentration (MEC_10_) of AEP, AET and AEE (µg/mL) on *E. coli* (**A**), *S. aureus* (**B**), *B. subtilis* (**C**), *P. aeruginosa* (**D**), and *S. pyogenes* (**E**). Six independent experiments, each with three technical replicates, compared to vancomycin (VAN) were considered (mean ± SD).

**Figure 6 molecules-25-02390-f006:**
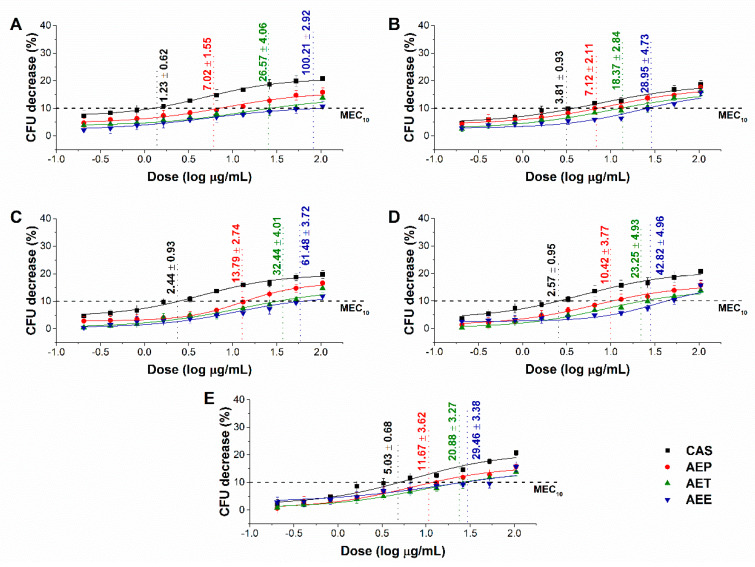
Minimum effective concentration (MEC_10_) of AEP, AET, and AEE (µg/mL) on *S. pombe* (**A**), *C. albicans* (**B**), *C. tropicalis* (**C**), *C. dubliniensis* (**D**) and *C. krusei* (**E**). Six independent experiments, each with three technical replicates, compared to caspofungin (CAS) were considered (mean ± SD).

**Figure 7 molecules-25-02390-f007:**
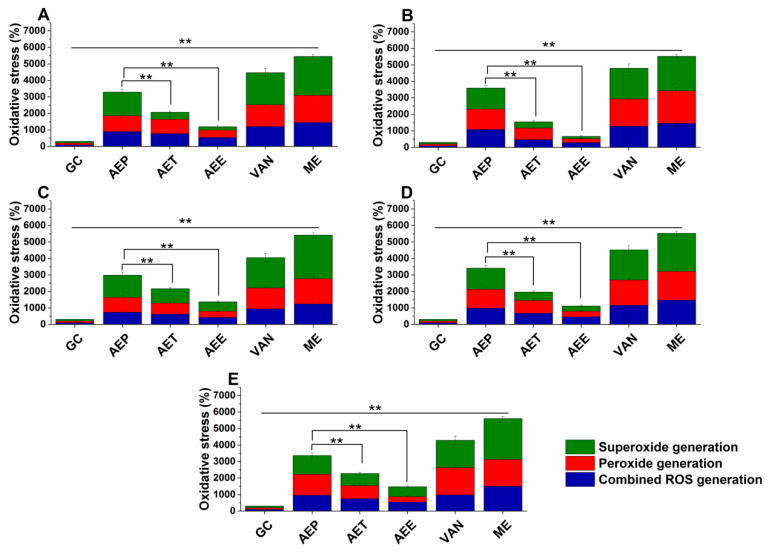
Percentage oxidative stress generation by AEP, AET, AEE, and VAN on *E. coli* (**A**), *S. aureus* (**B**), *B. subtilis* (**C**), *P. aeruginosa* (**D**), and *S. pyogenes* (**E**) at their respective MEC_10_ concentrations after one-hour treatment. All results were compared to those for menadione (ME) and growth control (GC). The three formulations were evaluated separately as well. Six independent experiments, each with three technical replicates (** *P* < 0.01, mean ± SD).

**Figure 8 molecules-25-02390-f008:**
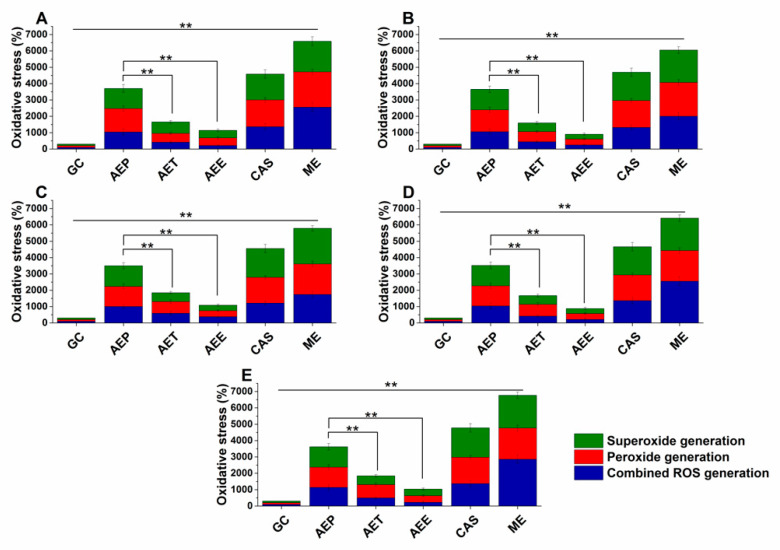
Percentage oxidative stress generation by AEP, AET, AEE, and CAS on *S. pombe* (**A**), *C. albicans* (**B**), *C. tropicalis* (**C**), *C. dubliniensis* (**D**) and *C. krusei* (**E**) at their respective MEC_10_ concentrations and after one-hour treatment. All results were compared to those for menadione (ME) and growth control (GC). The three formulations were evaluated separately as well. Six independent experiments, each with three technical replicates (** *P* < 0.01, mean ± SD).

**Figure 9 molecules-25-02390-f009:**
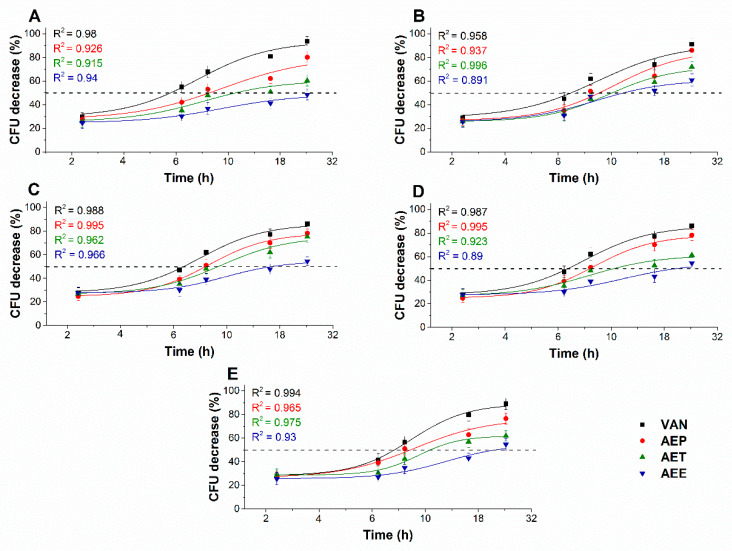
Effects of AEP, AET, and AEE at their MEC_10_ concentrations on the mean percentage colony-forming unit (CFU/mL) decrement of the planktonic *E. coli* (**A**), *S. aureus* (**B**), *B. subtilis* (**C**), *P. aeruginosa* (**D**), and *S. pyogenes* (**E**) compared to their respective 0 h samples and vancomycin (VAN) standard antimicrobial controls after 2, 6, 8, 16 and 24 h of treatment (mean ± SD, *n* = 6 independent experiments each with three technical replicates).

**Figure 10 molecules-25-02390-f010:**
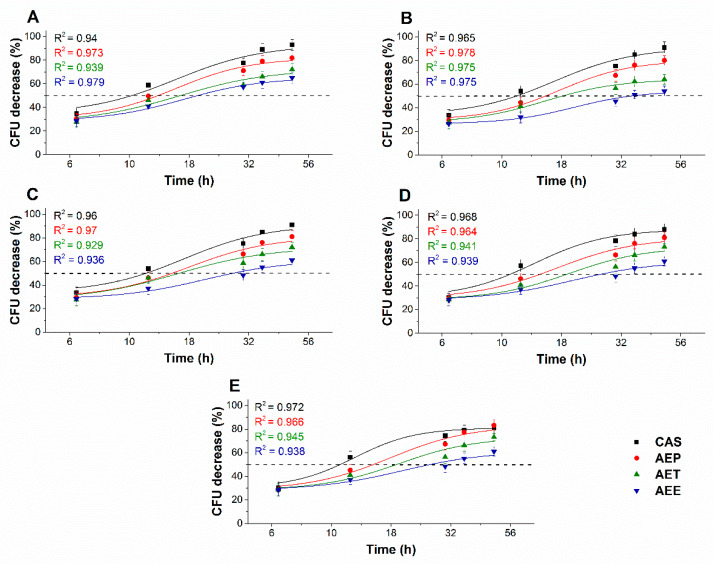
Effects of AEP, AET, and AEE at their MEC_10_ concentrations on the metabolic activities of the mean percentage colony-forming unit (CFU/mL) decrement of the planktonic *S. pombe* (**A**), *C. albicans* (**B**), *C. tropicalis* (**C**), *C. dubliniensis* (**D**) and *C. krusei* (**E**) compared to their respective 0 h samples and caspofungin (CAS) standard antimicrobial controls after 6, 12, 30, 36 and 48 h of treatment (mean ± SD, *n* = 6 independent experiments each with three technical replicates).

**Figure 11 molecules-25-02390-f011:**
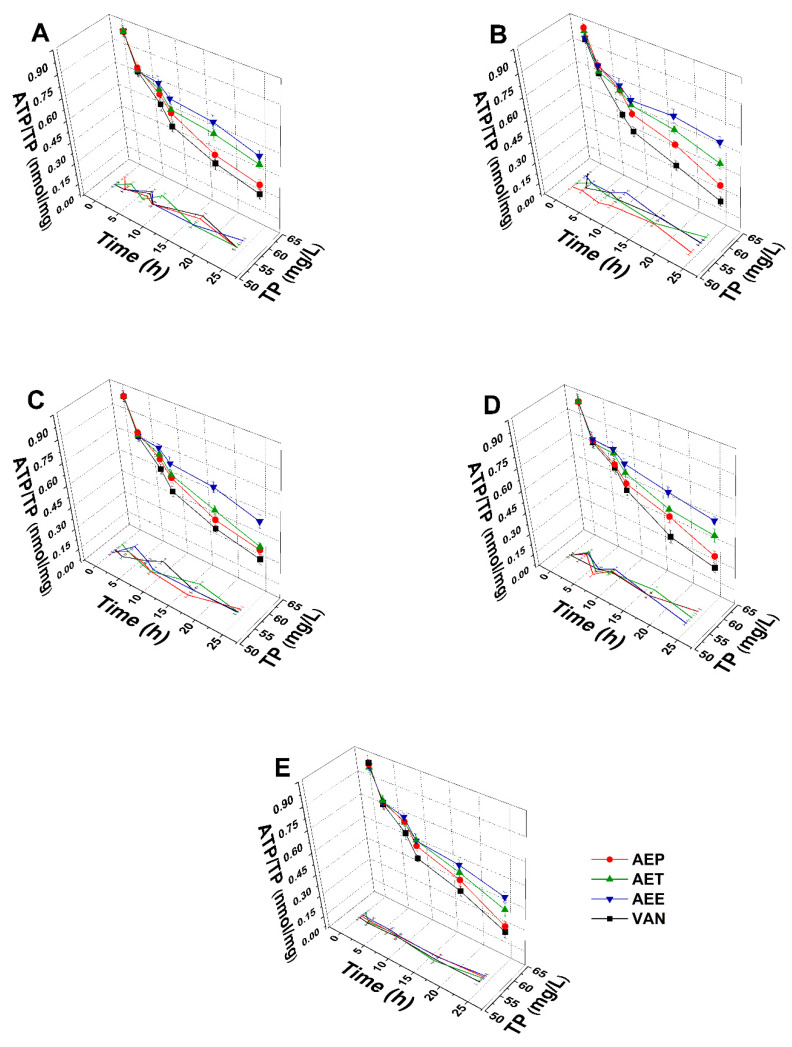
Effects of AEP, AET and AEE at their MEC_10_ concentrations on the total protein content (TP) and intracellular ATP/TP ratio of the planktonic *E. coli* (**A**), *S. aureus* (**B**), *B. subtilis* (**C**), *P. aeruginosa* (**D**), and *S. pyogenes* (**E**) compared to their respective 0 h samples and vancomycin (VAN) standard antimicrobial controls after 2, 6, 8, 16 and 24 h of treatment (mean ± SD, *n* = 6 independent experiments each with three technical replicates).

**Figure 12 molecules-25-02390-f012:**
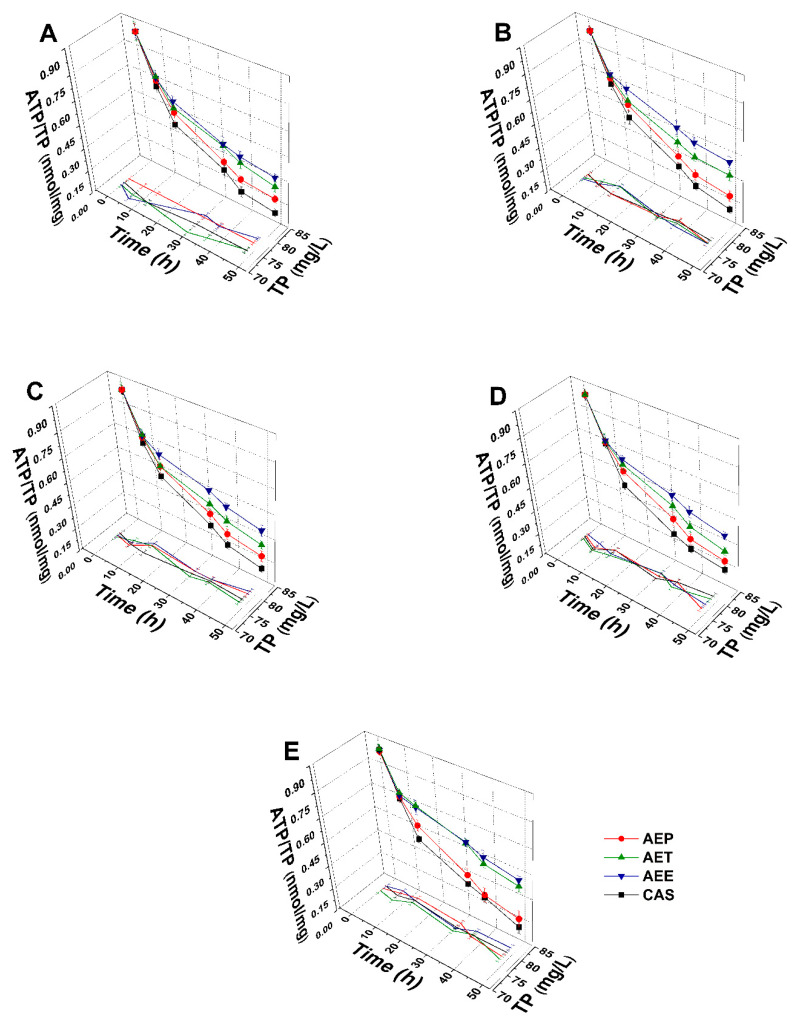
Effects of AEP, AET, and AEE at their MEC_10_ concentrations on the total protein content (TP) and intracellular ATP/TP ratio of the planktonic *S. pombe* (**A**), *C. albicans* (**B**), *C. tropicalis* (**C**), *C. dubliniensis* (**D**) and *C. krusei* (**E**) compared to their respective 0 h samples and caspofungin (CAS) standard antimicrobial controls after 6, 12, 30, 36 and 48 h of treatment (mean ± SD, *n* = 6 independent experiments each with three technical replicates).

**Figure 13 molecules-25-02390-f013:**
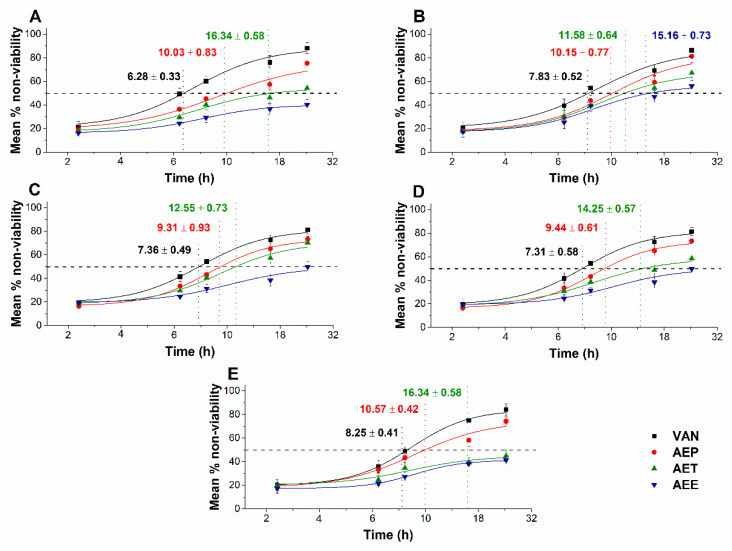
Mean percentage non-viability of AEP, AET and AEE at their MEC_10_ concentrations on the metabolic activities of the planktonic *E. coli* (**A**), *S. aureus* (**B**), *B. subtilis* (**C**), *P. aeruginosa* (**D**), and *S. pyogenes* (**E**) compared to their respective 0 h samples and vancomycin (VAN) standard antimicrobial controls after 2, 6, 8, 16 and 24 h of treatment (mean ± SD, *n* = 6 independent experiments each with three technical replicates).

**Figure 14 molecules-25-02390-f014:**
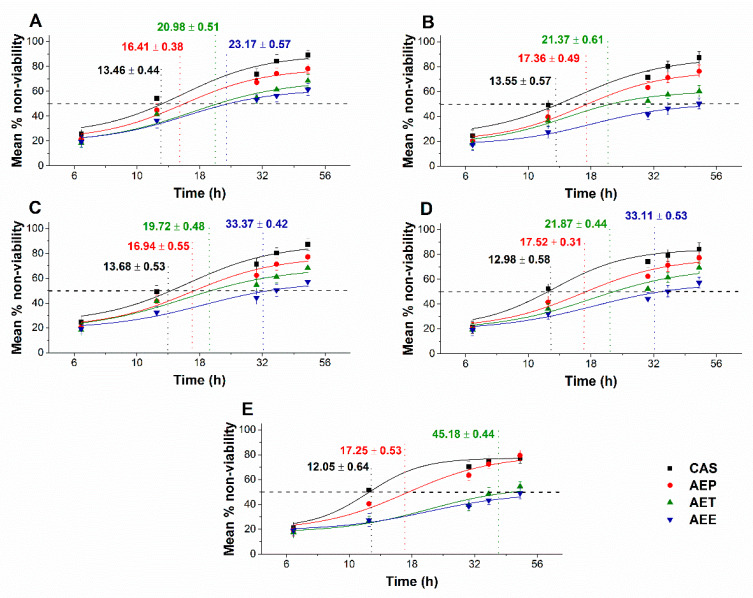
Mean percentage non-viability of AEP, AET, and AEE at their MEC_10_ concentrations on the metabolic activities of the planktonic *S. pombe* (**A**), *C. albicans* (**B**), *C. tropicalis* (**C**), *C. dubliniensis* (**D**) and *C. krusei* (**E**) compared to their respective 0 h samples prior and caspofungin (CAS) standard antimicrobial controls after 6, 12, 30, 36 and 48 h of treatment (mean ± SD, *n* = 6 independent experiments each with three technical replicates).

**Figure 15 molecules-25-02390-f015:**
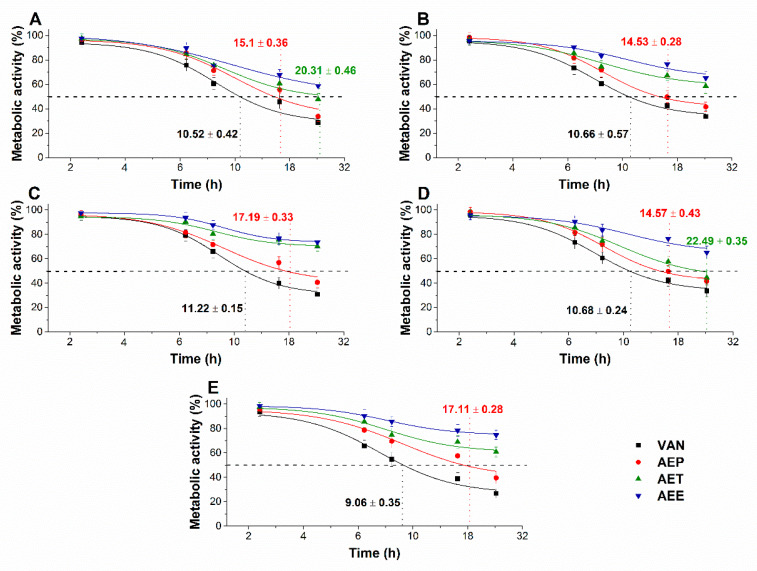
Effects of AEP, AET and AEE at their MEC_10_ concentrations on the metabolic activities of the planktonic *E. coli* (**A**), *S. aureus* (**B**), *B. subtilis* (**C**), *P. aeruginosa* (**D**), and *S. pyogenes* (**E**) compared to their respective 0 h samples and vancomycin (VAN) standard antimicrobial controls after 2, 6, 8, 16 and 24 h of treatment (mean ± SD, *n* = 6 independent experiments each with three technical replicates).

**Figure 16 molecules-25-02390-f016:**
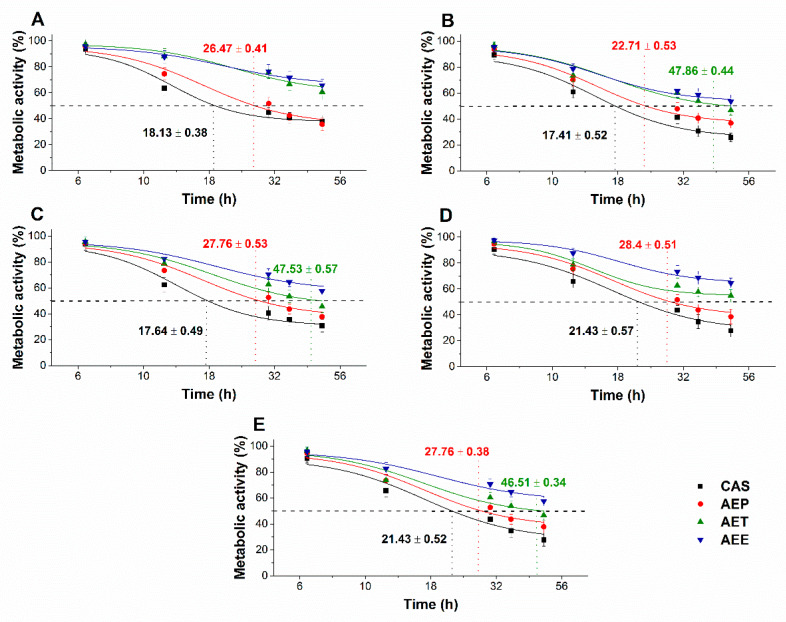
Effects of AEP, AET and AEE at their MEC_10_ concentrations on the metabolic activities of the planktonic *S. pombe* (**A**), *C. albicans* (**B**), *C. tropicalis* (**C**), *C. dubliniensis* (**D**) and *C. krusei* (**E**) compared to their respective 0 h samples and caspofungin (CAS) standard antimicrobial controls after 6, 12, 30, 36 and 48 h of treatment (mean ± SD, *n* = 6 independent experiments, each with three technical replicates).

**Figure 17 molecules-25-02390-f017:**
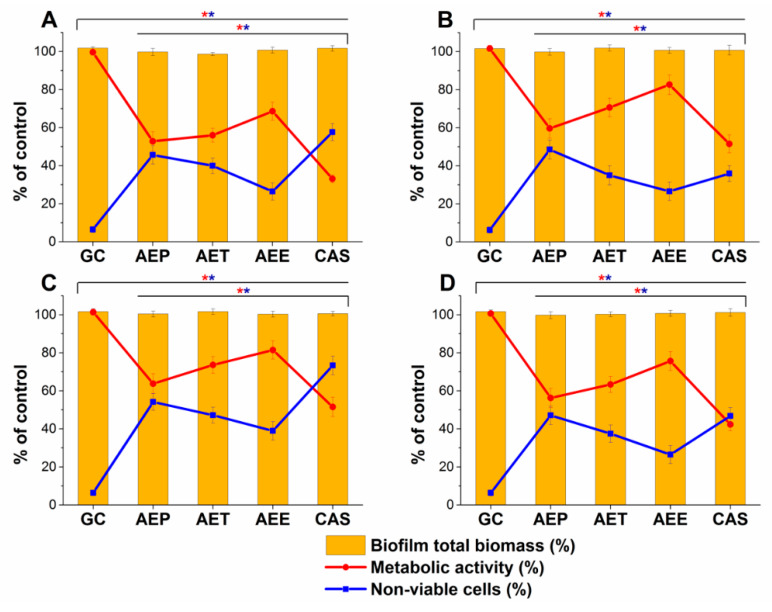
Effects of AEP, AET and AEE after 24 h of treatments at their respective MEC_10_ concentrations on the metabolic activity, amount of biofilm biomass, and viability of *C. albicans* (**A**), *C. tropicalis* (**B**), *C. dubliniensis* (**C**) and *C. krusei* (**D**) cell populations (mean ± SD, *n* = 6 independent experiments each with three technical replicates, data were compared with untreated controls (GC) and with caspofungin (CAS)-treated positive controls. The red (*****) and blue (*****) asterisks represent a significance value of *P < 0.01* for the metabolic activity and viability measurements, respectively.

**Figure 18 molecules-25-02390-f018:**
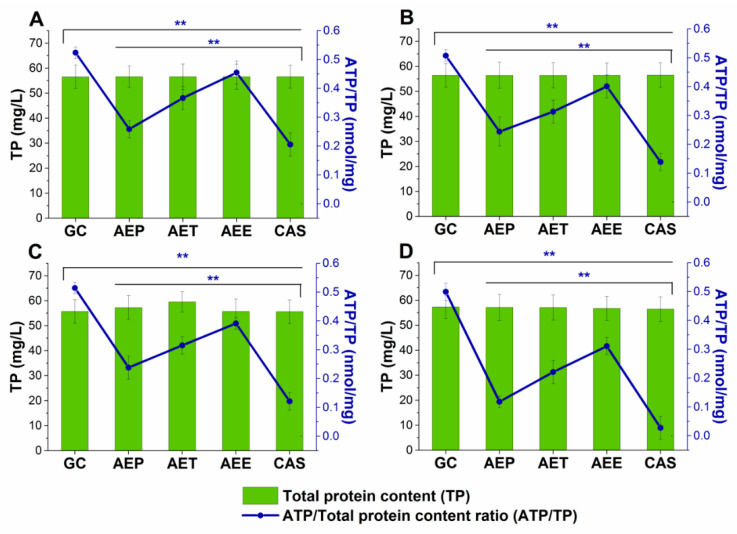
Effects of AEP, AET and AEE after 24 h of treatments at their respective MEC_10_ concentrations on the ATP and the total protein content (TP) of *C. albicans* (**A**), *C. tropicalis* (**B**), *C. dubliniensis* (**C**) and *C. krusei* (**D**) cell populations in the mature biofilms (mean ± SD, *n* = 6 independent experiments each with three technical replicates, data were compared with untreated controls (GC) and with caspofungin (CAS)-treated positive controls. The blue double asterisks (******) represent a significance value of *P*
*< 0.01* for the intracellular ATP/Total protein content (ATP/TP), respectively.

**Table 1 molecules-25-02390-t001:** Stability parameters and droplet sizes of Pickering nanoemulsion and conventional emulsion.

Stabilizing Agent	c_oil_ (mg/mL)	D_droplet_ (nm) ± SD	Stability
SNP	0.2–3.5	160 ± 2.2–670 ± 37.2	2–3 months
Tw80	0.2–3.5	130 ± 0.9–590 ± 19.6	2–3 months

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
