# Peer review of "Antimicrobial Activity of Different Artemisia Essential Oil Formulations"

_molecules, 2020, doi:10.3390/molecules25102390_

Round 1

Reviewer 1 Report

I reviewed the manuscript entitled Title: “Antimicrobial Activity of Different Artemisia

Essential Oil Formulations ".

General comments:

This is an interesting research work related to the “Antimicrobial activity of Artemisia Essential oils formulations”  including  the viability and cytotoxic effects also the comparison of different formulations. The paper presents a comprehensive study described in a clear straightforward and detailed form. Also, the manuscript is well-written. To my opinion, it can be published after just minor technical improvement.

Abstract. Ok

Introduction. OK

Results

Line 255-266: The effect on viability and metabolic activity are mentioned but the relationship of these tests is not mentioned as part of a citotoxicity,  since in the method section it comes as cytotoxicity. Please clarify. It would also be good to mention which method was used for each determination (Resarzurine or Sybr Green) since although it is mentioned in methodology, no mention is made at any time in the results and creates confusion due to the length of the work.

Discussion: OK

Material and methods.

Line 419-436: Among the materials and methods, Mueller Hinton, Yeast Extract and Peptone are mentioned, which are not mentioned in the experiments. remove from material and method or clarify what they were used for.

Line 429: correct “Me”  (menadione) for “ME”. In all the document appears like ME and causes confusion.

Line 461: Mention the previous protocol but the reference is not mentioned.

Line 601-611: Does not mention how the crystal violet staining was performed, clarify.

References:

Line 710: The name of thejournal is missing.

Reviewer 2 Report

Comments and suggestions for authors are attached. 

Round 2

Reviewer 2 Report

Line 257

Delete the word “dramatic”. It’s not used in technical/scientific writing.  

Lines 770-771

“The collected samples were stabilized on ice for 5 minutes, followed by 5 minutes vortexing and a rest period on ice simultaneously.”

How much is the rest period on ice? What do the authors mean by “simultaneously”? It’s confusing. The sentence may be rephrased!  

Line 771

Replace “minutes, followed by 5 minutes” with “min, followed by 5 min”  
